# Analysis of the Composition of *Deinagkistrodon acutus* Snake Venom Based on Proteomics, and Its Antithrombotic Activity and Toxicity Studies

**DOI:** 10.3390/molecules27072229

**Published:** 2022-03-29

**Authors:** Jin Huang, Minrui Zhao, Chu Xue, Jiqiang Liang, Fang Huang

**Affiliations:** 1Shenzhen Pingle Orthopedic Hospital (Shenzhen Pingshan Traditional Chinese Medicine Hospital), Shenzhen 518001, China; cpuhuangjin@163.com (J.H.); 15601912979@163.com (M.Z.); 2Jiangsu Key Laboratory of TCM Evaluation and Translational Research, Department of Pharmacology of Chinese Materia Medica, China Pharmaceutical University, 639 Longmian Road, Nanjing 211198, China; xuechu0415@163.com

**Keywords:** *Deinagkistrodon acutus* venom, proteomic analysis, label-free proteomics, hematotoxicity, antithrombosis, anticoagulation

## Abstract

There is a strong correlation between the composition of *Deinagkistrodon acutus* venom proteins and their potential pharmacological effects. The proteomic analysis revealed 103 proteins identified through label-free proteomics from 30 different snake venom families. Phospholipase A2 (30.0%), snaclec (21.0%), antithrombin (17.8%), thrombin (8.1%) and metalloproteinases (4.2%) were the most abundant proteins. The main toxicity of *Deinagkistrodon acutus* venom is hematotoxicity and neurotoxicity, and it acts on the lung. *Deinagkistrodon acutus* venom may have anticoagulant and antithrombotic effects. In summary, the protein profile and related toxicity and pharmacological activity of *Deinagkistrodon acutus* venom from southwest China were put forward for the first time. In addition, we revealed the relationship between the main toxicity, pharmacological effects, and the protein components of snake venom.

## 1. Introduction

Venom toxins represent a vast untapped source of pharmacologically-active lead compounds for drug development [1]. Snake venom is a highly complex mixture containing abundant proteins and polypeptides with diverse biological activities [2]. Recent research indicates that snake venom contains many active proteins or peptides with defibration, anticoagulation, antiplatelet aggregation, and antithrombotic functions [3]. A large number of novel snake venom-related preparations have been widely used in clinical practice. Effective separation and purification are crucial steps toward discovering novel snake venom proteins. Chromatographic separation has been a traditional method for identifying the chemical constituents of snake venom, however, it is unable to uncover the holistic proteome profile of the venom [4]. The protein composition of snake venom is the difficulty and key to snake venom research. Novel proteomic techniques provide an opportunity to study the composition of snake venom proteins by combining gel electrophoresis with LC-MS. Proteomics technology can comprehensively analyze the composition of the overall proteome in venom and provide direction and data reference for researching new snake venom components. In addition to discovering novel snake venom protein, the detailed characterization of the venom proteome will allow us to deeply understand the pathogenesis and biological function of snake envenomation [5]. With the emergence of omics techniques, the field of proteomics has vastly expanded due to the availability of large sets of proteomic and genomic databases resulting in the detailed elucidation of the composition of many snake venom [6,7]. It is worth noting that the proteome profiles of the venoms of the Asiatic non-spitting cobra, N. naja atra [8], and Pakistan N. naja [9] were poorly understood.

*Deinagkistrodon acutus* is a medically necessary snake in many South Asian countries, including China [10]. The venom of the *Deinagkistrodon acutus* primarily leads to coagulopathy and promotes tissue damage, edema, necrosis, hemostatic imbalance, hemorrhage, and acute renal failure [11]. The researchers studied the protein composition of the snake venom by different technical methods and found new snake venom proteins. Transcriptome analysis of *Deinagkistrodon acutus* venomous glands uses expressed sequence tags to focus on cellular structural and functional aspects [12]. Zhang’s group built a research platform for snake venom gland transcriptome sequencing and snake venom functional gene mining [13]. In 2016, the first transcriptome analysis of the *Deinagkistrodon acutus* venom gland reported a 1.43 Gb genome and tissue transcriptomes (including venom gland). This study annotated a total of 35 venom genes or gene families using all the known snake venom proteins [14]. However, these studies did not directly analyze the protein composition of the snake venom. Fortunately, based on proteomics, researchers found that snake venom metalloproteinases and C-type lectins were the dominant proteins (84.45%) in *Deinagkistrodon acutus* venom in Taiwan of China [10]. Moreover, they explored the *Deinagkistrodon acutus* venom proteome by integrating a combinatorial peptide ligand library approach with shotgun LC-MS/MS [15]. The results indicated that the venom comprised 84 distinct proteins from 10 toxin families and 12 other proteins. The whole venom or some of the purified components of the *Deinagkistrodon acutus* venom have been subjected to biochemical and pharmacological characterization, such as MSP, Pt-A (Glu-Asn-Trp), SLPC, ACH-11 [16,17,18,19]. However, the protein compositions of *Deinagkistrodon acutus*’s venom in different regions may be different [20]. To better understand the pathophysiology, details about the venom of *Deinagkistrodon acutus* in the Chinese mainland still need to be explored. Based on the Label-free proteomics method, this study explored the compositions and contents of protein peptides in *Deinagkistrodon acutus* venom originating in the Chinese mainland. Based on proteomics results and literature, this study analyzed the primary toxicity and antithrombotic activity of the *Deinagkistrodon acutus* venom. It provides the basis for new drug research and specific anti-snake venom serum [21].

## 2. Material and Methods

### 2.1. Materials Preparation

For experimental proteomics analysis, venom powder from *Deinagkistrodon acutus* was purchased from Xiangtan Snake farm (Xiangtan, China) and stored at −80 °C till use.

Adult male ICR mice (18–22 g, SPF grade) were purchased from the Comparative Medical Center of Yangzhou University (Yangzhou, China, License No. SYXK (su) 2018-0019). Adult male Sprague-Dawley rats (180 ± 20 g, SPF grade) were supplied by the Laboratory Animal Center of Nanjing Qinglongshan (Nanjing, China, License No. SYXK-Su-2018-0047). The operation of animal experiments conformed to the Guideline of the Ethical Committee of the China Pharmaceutical University. The ethics approval number was AUC-37 (20180329).

### 2.2. Venom Proteome Analysis

The protein concentration was determined by the Bradford Protein Concentration Assay Kit (Thermo Fisher Scientific Incorporation, Santa Clara, CA, USA). Protein fractions were separated by 15% SDS-PAGE (reducing conditions) and stained with Coomassie Blue R-250. The enzymatic hydrolysis of the protein is referred to by Katayama H, Nagasu T, and Oda Y [22]. Images were acquired by Bio-Rad and quantified by densitometry analysis using Image J software.

#### 2.2.1. Nano-Liquid Chromatography–Tandem Mass Spectrometry Analysis

Peptides in 0.1% formic acid were loaded onto an EASY-nLC1200 nano-liquid chromatography (LC) system (Thermo Fisher Scientific, San Jose, CA, USA) equipped with a C18 PepMap trap column (750 μm ID × 150 mm, Thermo Fisher Scientific). Liquid phase liquid A is 0.1% formic acid-water solution, and liquid B is 0.1% formic acid-acetonitrile solution. The peptides were eluted from the trap column and separated using 0.1% formic acid in acetonitrile at a flow rate of 300 mL/min on an RP-C18 Tip column (75 μm ID × 150 mm, Thermo Fisher Scientific) with a spray voltage of 1.8 kV. Peptides were separated on a 0–55% acetonitrile gradient (90 min) with 0.1% formic acid at a 300 nL/min flow rate. The peptide ions in the spray were detected and analyzed on a Q-Exactive mass spectrometer (Thermo Fisher Scientific, Bremen, Germany), which was operated in the positive mode to measure full scan MS spectra (from *m*/*z* 300–1800 in the Q-Exactive analyzer MS^1^ at resolution R = 70,000, MS^2^ at resolution R = 17,500) followed by isolation and fragmentation of the 10 most intense ions (MS^2^ scan) (in the Q-Exactive part) by collision-induced dissociation. The normalized collision energy was 35.0. Dynamic exclusion was employed within 30 s to prevent the repetitive selection of peptides. Skyline software calculated the relative contents of protein in snake venom based on the iBAQ value of ionic strength.

#### 2.2.2. Protein Identification

The raw mass spectrometry MS/MS spectra search was processed using Mascot 2.3-associated Andromeda search engine (Matrix Science) and a snake peptide database (44,716 sequences) obtained from NCBI (NCBI_sequence_snak.fasta) (https://www.ncbi.nlm.nih.gov/protein?term=txid36307[Organism], accessed on 11 March 2021) in 11 March 2021. Initial maximum precursor and fragment mass deviations were set to 30 ppm and 0.15 Da, respectively. Variable modification (Acetyl (Protein N-term), Deamidated (NQ), Deoxidation (W), Oxidation (M)) and fixed modification (Carbamidomethyl (C)) were set for the search, and trypsin with a maximum of two missed cleavages was chosen for searching. The minimum peptide length was set to seven amino acids, and the false discovery rate (FDR) for peptide and protein identification was set to 0.05. The precursor ion mass accuracy was improved using the time- and mass-dependent recalibration option software. The false discovery rate was controlled at various levels using a target-decoy search strategy, which integrates multiple peptide parameters, such as length, charge, number of modifications, and the identification score into a single quality that acts as statistical evidence of the quality of each single peptide spectrum match. The frequently observed laboratory contaminants were removed, and the protein identification was considered valid only when at least one matched peptide and one unique peptide were present (confidence value ≥ 95%).

#### 2.2.3. Protein Function and Interaction Analyses

In 11 March 2021, UniProt (https://www.uniprot.org/, accessed on 11 March 2021), database David 6.7 (http://david.abcc.ncifcrf.gov/, accessed on 11 March 2021) and Cluster of Orthologous Groups of proteins (https: //www.ncbi.nlm.nih.gov/COG/, accessed on 11 March 2021) were performed to explain potential biological properties of identified proteins. Pathway mapping of identified proteins was performed using the KEGG database (http://www.genome.jp/kegg/, accessed on 11 March 2021) [23]. Protein-protein interaction was analyzed using the STRING database [24].

### 2.3. Lethal Activity of Deinagkistrodon acutus Venom

The lethality of the whole venom or isolated toxins was tested by intravenous (i.v.) injection in SPF grade ICR mice (18–22 g). Different concentrations of venom (0–5.0 mg/kg) were dissolved in 0.9% NaCl solution. Various doses of venom were injected in the caudal vein, deaths occurring within 24 h were recorded, and the LD_50_ values were calculated by probits. Tissue samples were collected from the mice immediately after death and placed in a formalin fixative solution. After routine processing, samples were embedded in paraffin, and the sections were prepared and stained with hematoxylin–eosin for histological analysis.

### 2.4. In Vitro Platelet Aggregation Assay

The PRP and PRP were harvested from SD rats. An aliquot of 8 mL blood was gently mixed with 880 µL citrate sodium anticoagulant (3.8%, *v*/*v*). Afterward, platelet-rich plasma (PRP) was obtained after centrifugation at 1000 rpm/min for 10 min. The residue was further centrifuged at 3000× *g* rpm/min for 10 min to obtain platelet-poor plasma (PPP). *Deinagkistrodon acutus* venom solution (0.5 mg/ mL) was preincubated with PRP at 37 °C for 5 min and then stimulated with different aggregating agents at the following final concentrations (ADP 15 μM or normal saline). Platelet aggregation was assayed by a platelet aggregation instrument (AggRAM, Helena, MT, USA), and the maximum aggregation rate was recorded in 5 min.

### 2.5. Anti-Coagulant Activity

#### Blood Clotting Assays

*Deinagkistrodon acutus* venom was injected into mice via the tail vein. Half an hour, coagulation time was measured by the capillary technique and slide method based on a previously published protocol. Blood samples were collected from the orbital plexus of mice. Afterward, the time for the blood coagulation was recorded. The clotting time was observed when microfibrils were visible. The average time of each observation was 20 s. The collected blood samples were also spotted on glass slides and clotting times were determined. Clotting was achieved when the fibrin mesh appeared and could be picked by a glass dissecting needle [25].

### 2.6. Collagen-Adrenalin-Induced Acute Pulmonary Thrombosis in Mice

After *Deinagkistrodon acutus* venom, normal saline and Enoxaparin sodium treatment for 1h, Collagen-adrenalin solution was intravenously administered to induce acute pulmonary embolism [26], which probably caused the paralysis or unpredicted death of mice. The recovery time of mice was recorded.

### 2.7. Carrageenan-Induced Tail Thrombosis in Mice

One hour after intravenous administration of *Deinagkistrodon acutus* venom, normal saline, and heparin sodium in each group, carrageenan solution (20 mg/kg) was intraperitoneally injected to induce tail thrombosis [27]. All mice were placed in a room at 20 °C for 12 h and the size of thrombosis in the tail was inspected. The length, pathological degree, and ratio of wine-colored tail thrombus were recorded.

### 2.8. Flow Restriction of the Inferior Vena Cava (IVC) in Mouse Induced Thrombosis

After *Deinagkistrodon acutus* venom and enoxaparin sodium treatment for 1 h. flow restriction of the inferior vena cava (IVC) in mice induced thrombosis was observed [20]. After 24 h, mice were anesthetized, and the IVC was excised below the ligation and proximal to the confluence of the common iliac vein. At harvest time, the IVC and the associated thrombus were removed from anesthetized animals. The weight and the thrombus length were recorded and used as a reference for thrombosis [28,29].

### 2.9. Statistical Analysis

The data were analyzed by SPSS 21.0 statistical software (IBM, Armonk, NY, USA) and presented as mean ± standard deviation (SD). One-way ANOVA analysis was used to evaluate the differences between different groups. A value of *p* < 0.05 was considered statistically significant.

## 3. Results

### 3.1. Proteomic Analysis of the Venom of Deinagkistrodon Acutus

First, sodium dodecyl sulfate-polyacrylamide gel electrophoresis (SDS-PAGE) was used to observe the molecular weight distribution of the snake venom protein (Figure 1). The molecular weight of the venom proteins ranged from 14.4 to 116 kDa. The predominant proteins were at 60 kDa, 20–26 kDa, and 14.4 kDa (Figure 1). Then, the protein bands were analyzed by label-free proteomics. The value of peptide parent ion mass tolerance is between −0.1 to 0.1 Da (Appendix A). The length and the number of unique peptides were determined to estimate the quality of the proteomic data (Appendix A). Based on the qualified data, 1407 peptides, matching 103 proteins, were identified in the snake venom (Figure 1). These identified proteins can be divided into two main groups, 42 enzymatic (32%) and 61 non-enzymatic proteins (68%) (Figure 2A). The predominant enzymatic proteins include phospholipase A2 acidic precursor (13.803%), snaclec agkicetin-C subunit α (12.559%), antithrombin A B-chain (12.478%), Lys-49 Phospholipase A2 precursor (10.743%), thrombin-like protein acutobin precursor (7.159%), and antithrombin AB-chain (2.478%). The following constituted the most abundant non-enzymatic proteins, snaclec anticoagulant protein subunit A (3.888%), snaclec clone (3.437%), agkisasin-b (3.316%), agglucetin-α subunit precursor (1.608%), and agkisacutacin B-chain (1.555%) (Figure 2B). In summary, the venom’s proteome comprises phospholipase A2, snaclec agkicetin, antithrombin, and metalloproteinase as the most abundant constituents based on the iBAQ value (Table 1). 

UniProt, David 6.7, and Cluster of Orthologous Groups of proteins were performed to explain the potential biological properties of *Deinagkistrodon acutus* venom. The main functional proteins included binding (15%), inhibition of platelet aggregation (14%), fibrinogen hydrolysate (9%), anticoagulant (8%), binding catalytic activity (7%), catalytic activity (5%), inhibition of platelet adhesion (4%) and antithrombotic and thrombolytic activity (4%) (Figure 3). The above results suggested that snake venom may have anticoagulant activity, antithrombotic activity, and contribute to coagulation disorders.

Using the KEGG database, the 67 pathways associated with the highest number of proteins identified from *Deinagkistrodon acutus venom* are shown in Figure 4, and all pathways found are recorded in Appendix A. Global and overview maps are the most enriched pathway identified for venom. Other enriched pathways include Metabolism of cofactors, vitamins Nucleotide metabolism, Transport and catabolism and Signaling molecules and interaction. Discussions relating to the interpretation of enriched KEGG pathways can be found in Appendix A. In Figure 5, the protein-protein interactions of the *Deinagkistrodon acutus* venom (PPI) network is shown.

### 3.2. Acute Toxicity Test of Deinagkistrodon acutus Venom

There was a positive association between the mortality and concentration of *Deinagkistrodon acutus* venom. Most of the mice died within 4 h. When the venom dose decreased, the survival rate of mice increased, the total mortality decreased, the mortality within 4 h decreased, and the pathological symptoms improved. The fitted curve between snake venom dose and mortality was y = 0.56 ln(x) + 0.0301, R^2^ = 0.9902. According to the analysis of the Bliss method, the LD_50_ of *Deinagkistrodon acutus* venom was 2.37 ± 0.357 mg/kg.

The primary toxicity of *Deinagkistrodon acutus* venom is hematotoxicity and neurotoxicity, and it obviously acts on the lung. Mice injected i.v. with *Deinagkistrodon acutus* venom presented with evident manifestations of limb paralysis and respiratory difficulties. After injection of different doses of snake venom, mice have the symptoms of ulceration of tail injury, even tail necrosis (Figure 6A). Mice had sluggish symptoms, severe convulsions, and subsequently, death (Figure 6B), and symptoms of mouth nose bleeding (Figure 6C). Mouse pleural blood, with the lungs spread through dark red pathological injury (Figure 6B,D) was found by anatomical studies on the mice.

The results of the HE staining section showed that a large number of red blood cell fragments and inflammatory cell infiltration appeared in the tail (Figure 7 Tail). A small amount of inflammatory cell infiltration and red blood cell fragments could be seen in the heart and liver (Figure 7 Heart, Liver). In addition, apparent tissue damage, inflammatory cell infiltration, and red blood cell fragments were observed in spleen and kidney tissues (Figure 6 Spleen, Kidney). It is worth noting that the lungs of mice were the most damaged. The lung tissue’s structural integrity was destroyed, the internal structure was unclear, and some endothelial cells were lysed and necrotic. Large thrombi were observed in pulmonary vessels. Similar to in the lung and kidney, there was inflammatory cell infiltration and red blood cell fragments in the lung tissue space (Figure 7 Lung).

As shown in Figure 8, *Deinagkistrodon acutus* venom could induce platelet aggregation and deplete the content of platelets in plasma. *Deinagkistrodon acutus* venom induces platelet aggregation and reduces platelet content in plasma, which may be important reasons for its pathological symptoms, such as thrombosis and coagulation disorders in mice.

### 3.3. Coagulant Activity In Vivo

Acute toxicity studies have shown that the primary toxicities of *Deinagkistrodon acutus* venom were hemotoxic and neurotoxic. *Deinagkistrodon*
*acutus* venom significantly prolonged clotting time and showed a positive dose-response correlation. (Figure 9A,B).

Snake venom inhibited the formation of a tail thrombus induced by carrageenan (Figure 10A). It also significantly inhibited acute pulmonary embolism formation induced by collagen-adrenalin. Compared to the control group, recovery time was significantly reduced, and recovery percentage was increased (Figure 10B). Snake venom inhibited the length and weight of thrombus in IVC mice (Figure 10C,D).

## 4. Discussion

The venom of *Deinagkistrodon acutus* is complex in composition and subject to geographical and ontogenetic variations [20]. Knowledge of the exact venom composition is, therefore, essential in selecting specimens to develop broad-spectrum polyvalent antiserum and clinical management of snake bites. Unlike *Deinagkistrodon acutus* venom from Taiwan, proteomic analysis revealed the presence of 29 distinct proteins from *D. acutus* venom belonging to eight snake venom protein families. Snake venom metalloproteinase (SVMP, 46.86%), C-type lectin (CLEC, 37.59%), phospholipase A2 (PLA2, 7.33%) and snake venom serine protease (SVSP, 6.62%) were the most abundant proteins [7]. The venom of the *Deinagkistrodon acutus* in the Chinese mainland comprises phospholipase A2, metalloproteinases, serine proteases, and snaclec proteins (Table 1). In these two studies, the origins of the venom of the *Deinagkistrodon acutus* are different, the analysis techniques and methods are also different, and the composition of the pit viper’s venom may be quite different. This study conducted a preliminary discussion on the toxicity and pharmacological activity of the venom of *Deinagkistrodon acutus*, which was lacking in Po-Cuan Chen’s research. Experiments on the toxicity and pharmacological activity of the venom of the *Deinagkistrodon acutus* provide a basis for clinical treatment. The pharmacological research direction of the snake venom monomer provides a data reference and basis as well as improves the research efficiency of snake venom and new snake venom monomer proteins. Previous studies mainly used RNA and other technical methods to determine the venom of *Deinagkistrodon acutus* in mainland China, but no scholars have directly determined the protein composition of *Deinagkistrodon acutus* venom by proteomic technology. This study determined the composition of the snake venom in mainland China and proved that there might be huge differences in the venom of the snake in mainland China and Taiwan due to differences in the regions, and the reasons for this difference deserve further exploration.

The venom protein composition of *Deinagkistrodon acutus* is closely related to its toxicity and antithrombotic and anticoagulant activity. The proteomic of the *Deinagkistrodon acutus* venom and literature analysis has shown that snake venom contains many active components that may affect blood coagulation and thromboses, such as antithrombin, thrombin-like, metalloproteinase, and serine proteinase [30,31]. We verified the main toxicity and potential pharmacological effects of *Deinagkistrodon acutus* venom through animal experiments.

The primary toxicity of *Deinagkistrodon acutus* venom is hematotoxicity and neurotoxicity, and it obviously acts on the lung. From a biological standpoint, it appears that prey immobilization and death induced by *Deinagkistrodon acutus* venom may occur by their different and highly effective mechanisms: intravascular thrombosis provoked by the prothrombin activator, thrombin and serine proteinase, blood coagulation disorders induced by the antithrombin, metalloproteinases and serine protease, and paralysis leading to respiratory failure induced by the PLA2. Observationally, mice developed toxic symptoms, such as pulmonary embolism, tail ulceration, and nose and mouth bleeding. The metalloproteinases obstruct homeostatic mechanisms, such as edema formation, inflammation, cell necrosis, and inflammation [30,32,33]. The action of prothrombin activators primarily results in defibrinogenation, factor depletion, and incoagulable blood, in some cases with active bleeding, instead of thrombosis [34,35,36]. The phospholipase A2 causes deterioration of skeletal muscles and inhibits neuromuscular transmission and the blood coagulation cascade [30,37]. The serine proteases cause hemostatic imbalances by affecting blood coagulation, fibrinolysis, and platelet function [33]. Phosphodiesterases are reported to possess potent anti-platelet aggregation potential [38,39].

The anticoagulant and antithrombotic effects of *Deinagkistrodon acutus* venom are related to many active protein components in the venom, such as phospholipase A2, metalloproteinases, serine proteases, antithrombin, thrombin. Pharmaceutical research shows that phospholipase A2 in snake venom may affect antiplatelet aggregation and affect platelet adhesion [39]. Metalloproteinases inhibit platelet aggregation, caused by inducers, such as ADP and collagen, and reduce platelet adhesion. Metalloproteinases can decompose fibrinogen, reduce fibrinogen content, and regulate the formation of thrombus and the spatial structure of a thrombus [18,40]. Some metalloproteinases play a thrombolytic effect by directly decomposing fibrin in the thrombus. Serine proteases regulate the activation of multiple coagulation factors in the coagulation cascade [41]. On the one hand, serine proteases can decompose fibrinogen, reduce fibrinogen content, inhibit platelet aggregation and thrombus formation [19,42]. On the other hand, serine proteases decompose fibrinogen to expose plasminogen binding sites, accelerate the activation rate of plasminogen, promote the activation of plasmin, and indirectly regulate the formation and dissolution of thrombus [43,44]. Thrombin-like enzymes decompose fibrinogen to produce soluble fibrin and regulate the formation of a thrombus. Soluble fibrin can be phagocytosed and decomposed by immune cells, such as macrophages, which cannot form an insoluble fibrin network to promote thrombus formation [44,45]. Antithrombin specifically binds to thrombin to reduce the activation rate of thrombin and inhibit thrombin from participating in the coagulation cascade, and finally plays an essential role in anticoagulant and antithrombotic effects [46].

In conclusion, the toxicity, anticoagulation, and antithrombotic effects of *Deinagkistrodon acutus* venom may be related to the active proteins in venom, such as phospholipase A2, metalloproteinase, serine protease, antithrombin, thrombin-like enzyme.

## 5. Conclusions

In this study, the composition and content of protein peptides in *Deinagkistrodon acutus* venom in mainland China was evaluated. Based on label free non-target quantitative analysis of *Deinagkistrodon acutus* venom protein, 103 proteins and 775 specific peptides were identified, and the relative content of various proteins in *Deinagkistrodon acutus* venom was determined. About 68% of the macromolecular snake venom protein is an enzyme protein, and 32% of the macromolecular snake venom proteins are non-enzymatic. The primary toxicity of *Deinagkistrodon acutus* snake venom is blood toxicity, and the main target organ is the lung. *Deinagkistrodon acutus* venom may have anticoagulant and antithrombotic effects, and its mechanism of action may be related to metalloproteases and serine proteases.

## Figures and Tables

**Figure 1 molecules-27-02229-f001:**
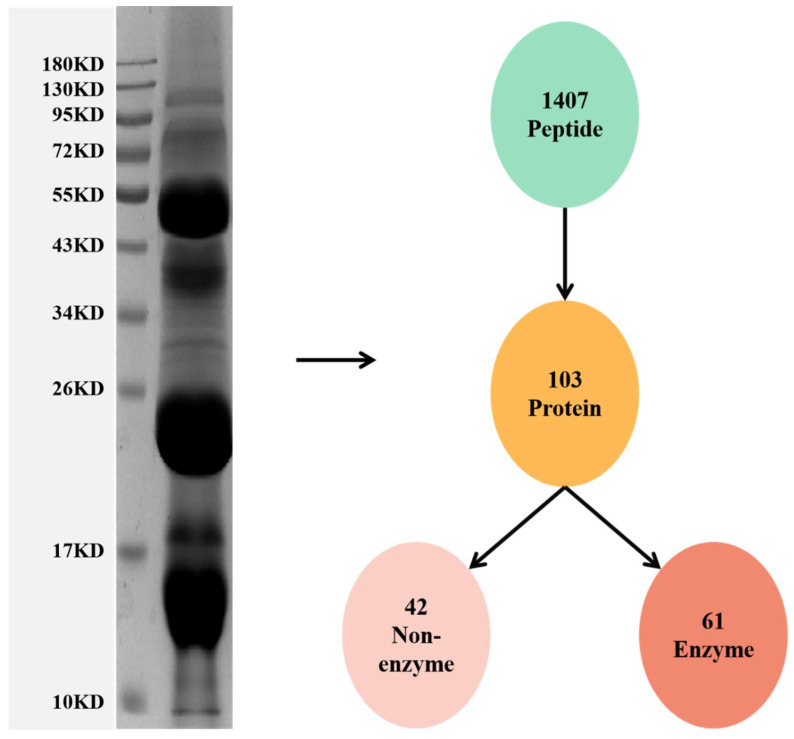
Proteomic analysis of *Deinagkistrodon acutus* venom. The venom powder was dissolved in lysis buffer and separated by SDS-PAGE. The protein bands were cut and subjected to protein identification by nano LC-MS/MS. The number of identified proteins was shown as a circle.

**Figure 2 molecules-27-02229-f002:**
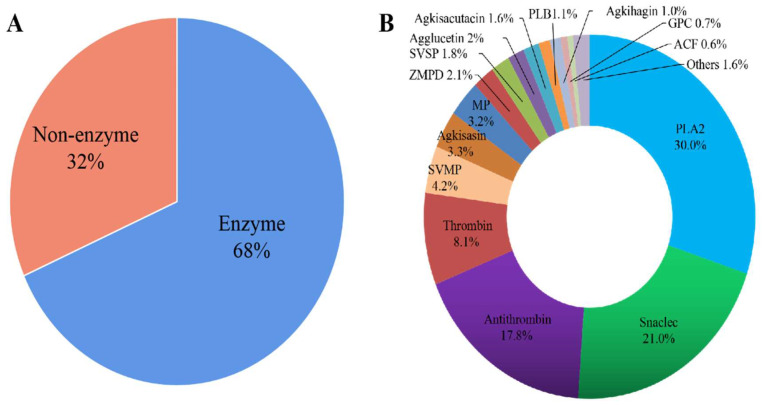
Overall protein composition of *Deinagkistrodon acutus* venom. (**A**) The classification of snake venom protein. (**B**) Overall protein composition of *Deinagkistrodon acutus*’s venom. The relative content of protein families in the venom of *Deinagkistrodon acutus*.

**Figure 3 molecules-27-02229-f003:**
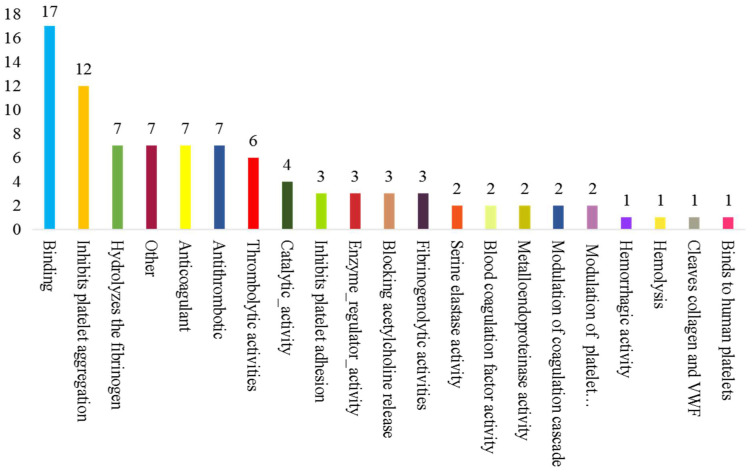
Molecular function of identified snake protein.

**Figure 4 molecules-27-02229-f004:**
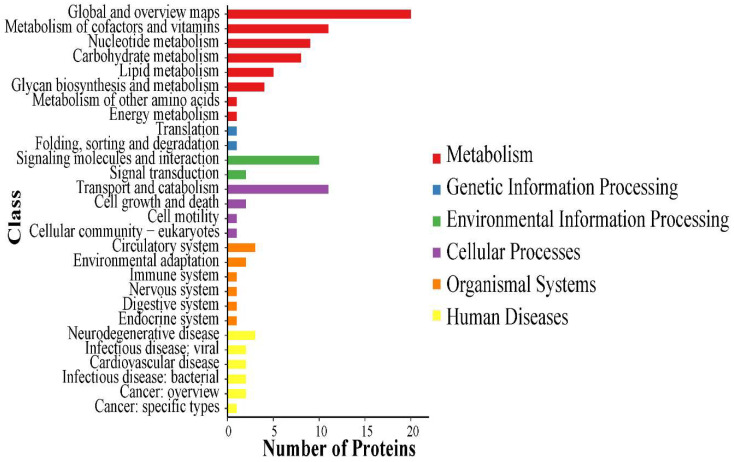
KEGG pathway classification. KEGG pathway. KEGG pathway includes Metabolism, Genetic Information Processin, Environmental Information Processing, Cellular Processes, Organismal Systems, Human Diseases.

**Figure 5 molecules-27-02229-f005:**
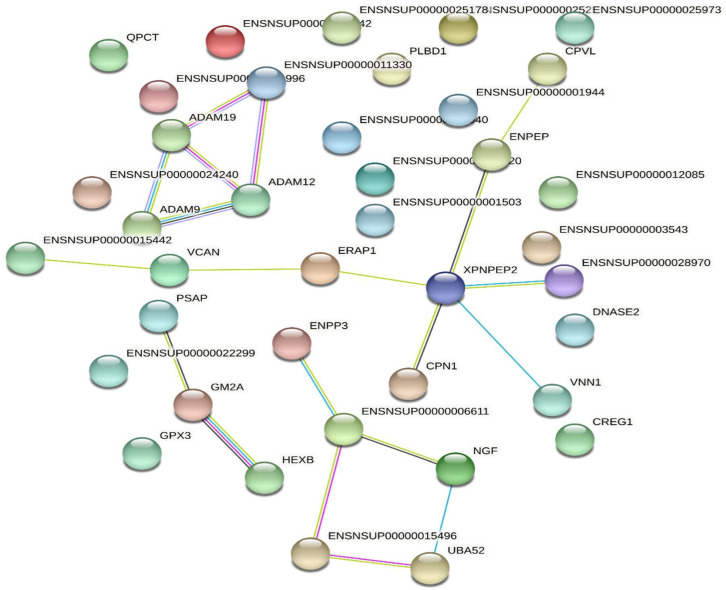
Protein-protein of Deinagkistrodon acutus venom interaction (PPI) network.

**Figure 6 molecules-27-02229-f006:**
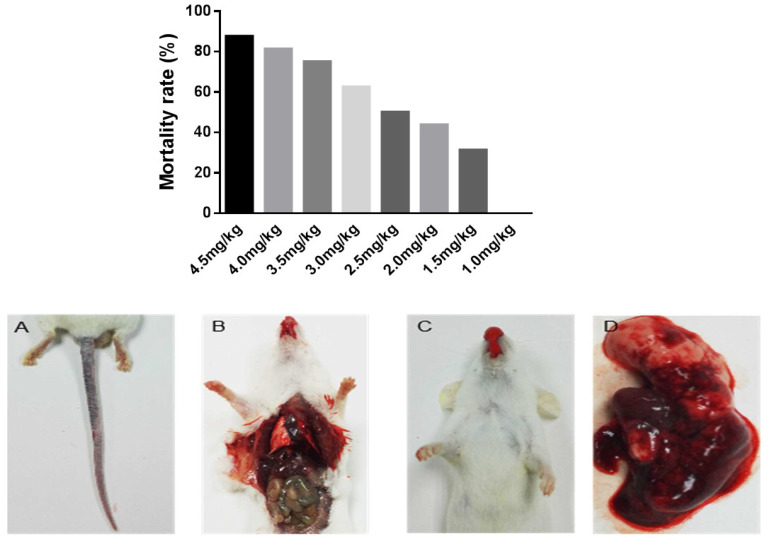
Acute toxicity test of *Deinagkistrodon acutus* venom. Acute toxicity death of *Deinagkistrodon acutus*. Pathological symptoms of mice with acute toxicity test of *Deinagkistrodon acutus*. (**A**) Tail (tail injury and purpura) (**B**) Peritoneal and thoracic cavity (internal hemorrhage) (**C**) Mouth nasal flow foamy blood (**D**) Lung (Pulmonary embolism and injury).

**Figure 7 molecules-27-02229-f007:**
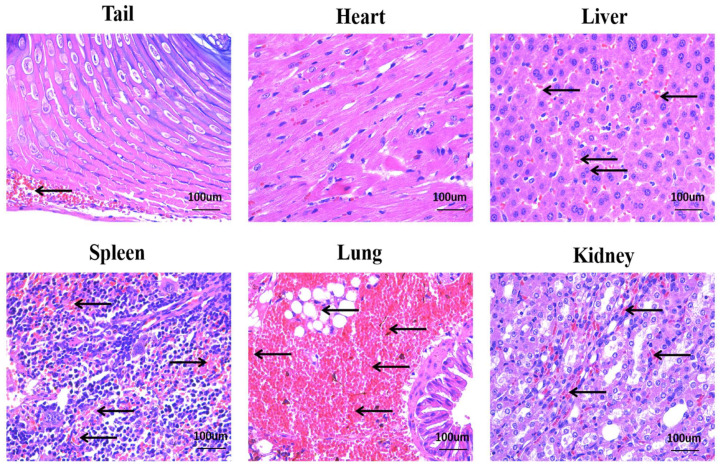
Toxicity analysis of *Deinagkistrodon acutus* venom. The effect of *Deinagkistrodon acutus* venom on mice. Inflammatory changes were observed in different groups using H & E staining (100 um). Arrowheads in these panels indicate inflammatory infiltration and damage.

**Figure 8 molecules-27-02229-f008:**
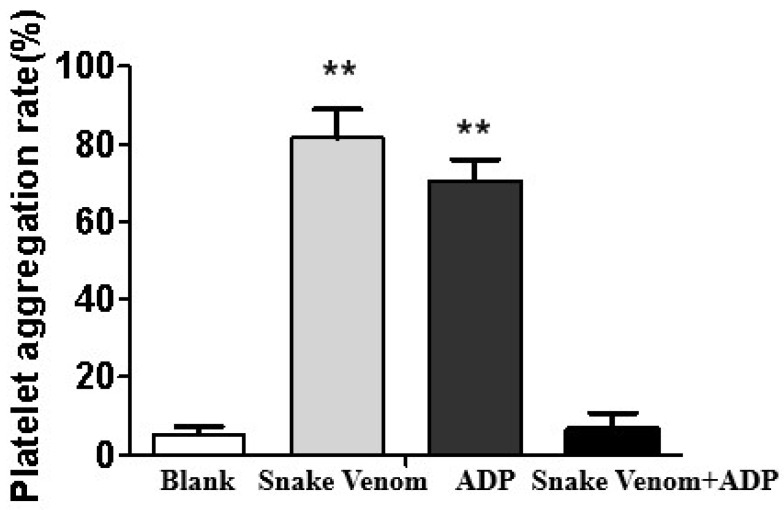
The effect of *Deinagkistrodon acutus* on platelet aggregation. All data were presented as mean ± SD, *n* = 10, ** *p* < 0.01, compared with the blank group.

**Figure 9 molecules-27-02229-f009:**
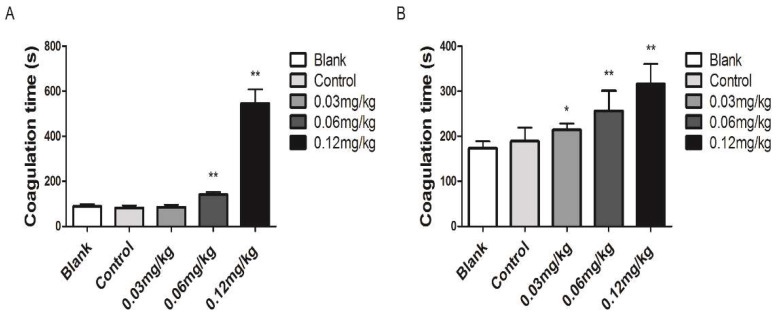
Anticoagulant activity of *Deinagkistrodon acutus* venom. (**A**) Blood coagulation time was determined using capillary methods. (**B**) Blood coagulation time was determined using the slide methods. All data are presented as mean ± SD, *n* = 10, * *p* < 0.05 and ** *p* < 0.01, compared with the control group.

**Figure 10 molecules-27-02229-f010:**
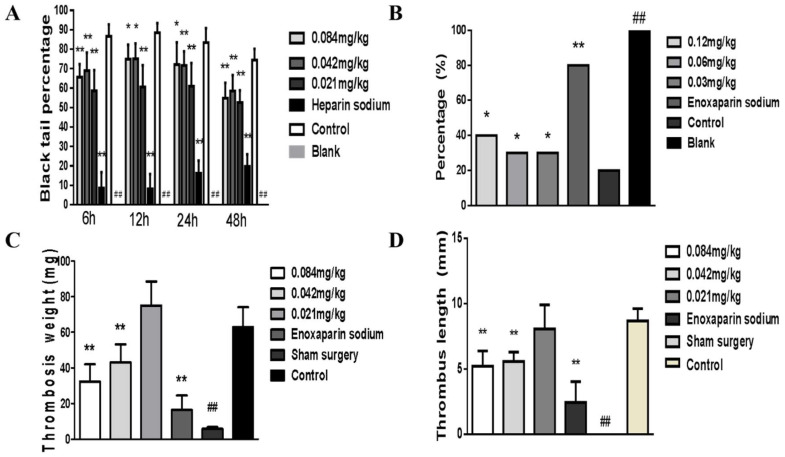
Effect of *Deinagkistrodon acutus* venom on thrombosis. (**A**) Effect of snake venom on carrageenan-induced tail thrombosis. (**B**) Effect of snake venom on collagen-adrenalin-induced acute pulmonary embolism. (**C**,**D**) Effect of snake venom on deep vein thrombosis induced by inferior vena cava ligation. The effect of snake venom on thrombosis was evaluated by measuring the length of blacktail induced by carrageenan, the survival rate of collagen-adrenalin-induced acute pulmonary embolism, and the length and weight of thrombus in IVC control. All data are presented as mean ± SD, *n* = 10, * *p* < 0.05, ** *p* < 0.01, compared to the control group.Blank control group and sham surgery group compared to the control group, all data are presented as mean ± SD, *n* = 10, ^##^
*p* < 0.01.

**Table 1 molecules-27-02229-t001:** Proteome profiles of *Deinagkistrodon acutus* venom.

Protein ID	Description	Coverage	Protein Mass	Peptide	Unique Peptide	Uniq Spectrum	snak_iBAQ	%	Function
CAJ85790.1	Phospholipase A2, acidic precursor, partial [*Deinagkistrodon acutus*]	0.9756	14,031.6	25	25	1191	6,623,486.35	13.80	Binding catalytic_activity, Inhibits platelet aggregation.
sp|Q9DEA2.1|SLCA_DEIAC	Snaclec agkicetin-C subunit alpha;	0.7935	17,798.4	30	9	259	6,026,939.81	12.56	Binding, Inhibits platelet aggregation, Inhibits platelet adhesion.
AAM22784.1	Antithrombin A B-chain [*Deinagkistrodon acutus*]	0.7671	16,688	22	22	678	5,987,824.836	12.48	Inhibits platelet aggregation, Inhibits platelet adhesion.
AAL36975.1	Lys-49 phospholipase A2 precursor [*Deinagkistrodon acutus*]	0.6739	15,803.4	35	35	550	5,154,993.67	10.74	Binding catalytic_activity, Indirect hemolytic activity and anticoagulant activity
AAF76377.1	Thrombin-like protein acutobin precursor [*Deinagkistrodon acutus*]	0.6923	28,814.8	29	25	917	3,435,511.45	7.16	Catalytic_activity, Coagulates human fibrinogen by hydrolysis of the alpha chains.
SPQ1ZY03.1|PA2B_DEIAC	Basic phospholipase A2 DAV-N6; Short = svPLA2; AltName: Full = Phosphatidylcholine 2-acylhydrolase; Flags: Precursor	0.7174	15,853.5	16	16	310	2,603,255.33	5.43	Binding catalytic_activity, Blocking acetylcholine release from the nerve termini.
AAL66390.1	Antithrombin 1 B chain [*Deinagkistrodon acutus*]	0.8591	17,249.8	41	15	274	2,102,366.73	4.38	Inhibits platelet aggregation, Inhibits platelet adhesion.
SP|Q9W7S2.1|VM11_DEIAC	Snake venom metalloproteinase aculysin-1; Short = SVMP; Flags: Precursor	0.4604	47,304.2	105	58	1689	1,918,894.51	4.00	Binding catalytic_activity, Hemorrhagic activity
SP|Q9DEF9.1|SLAA_DEIAC	Snaclec anticoagulant protein subunit A; Short = AaACP-A; Flags: Precursor	0.7829	17,124.2	27	19	1019	1,865,840.86	3.89	Binding, Inhibits platelet aggregation, Antithrombotic and thrombolytic activities, Hydrolyzes the fibrinogen.
SP|Q8JIV8.1|SL_DEIAC	Snaclec clone 2100755; AltName: Full = C-type lectin clone 2100755; Flags: Precursor	0.671	17,944.1	25	25	464	1,649,481.55	3.44	Binding, Modulation of platelet aggregation, or coagulation cascade.
AAK26430.1	Agkisasin-b [*Deinagkistrodon acutus*]	0.7823	14,701.3	25	3	29	1,590,986.26	3.32	Binding, Anticoagulant, Inhibits platelet aggregation, Antithrombotic and thrombolytic activities, Hydrolyzes the Aalpha-chain of fibrinogen.
AFR11355.1	Snake venom serine protease Da-36 [*Deinagkistrodon acutus*]	0.5654	29,057.1	17	12	110	874,312.48	1.82	Catalytic_activity, Blood coagulation factor activity, Serine elastase activity.
AAN23124.1	Agglucetin-alpha 1 subunit precursor [*Deinagkistrodon acutus*]	0.8052	17,317.4	33	3	405	771,737.66	1.61	Binding, Blood coagulation factor activity, Serine elastase activity.
AAM22785.1	Agkisacutacin B-chain [*Deinagkistrodon acutus*]	0.8288	16,739.8	37	7	488	746,482.39	1.56	Binding, Anticoagulant, Inhibits platelet aggregation, Antithrombotic and thrombolytic activities, Hydrolyzes the Aalpha-chain of fibrinogen.
BAO23490.1	Metalloproteinase [*Deinagkistrodon acutus*]	0.4745	67,764.7	48	41	702	740,656.11	1.54	Binding catalytic, Enzyme_regulator_activity, Metalloendoproteinase activity.
SP|Q9IAX6.1|VM2M2_DEIAC	Zinc metalloproteinase/disintegrin	0.6438	52,517.3	117	25	762	740,180.03	1.54	Binding, enzyme_regulator_activity, inhibits Platelet aggregation, Cleaves fibrinogen, Cleaves collagen and VWF.
PDB|3HDB|A	Chain A, Crystal Structure of Aahiv, A Metalloproteinase From Venom of *Deinagkistrodon acutus*	0.8345	46,681.5	95	12	590	639,917.60	1.33	Binding, Anticoagulant, Inhibits platelet aggregation, Antithrombotic and thrombolytic activities, Hydrolyzes the Aalpha-chain of fibrinogen.
SP|Q9IAM1.2|SLUA_DEIAC	Snaclec agkisacutacin subunit A; Short = Agk-A; Flags: Precursor	0.7895	17,109.1	21	3	287	502,397.74	1.05	Binding, Anticoagulant, Inhibits platelet aggregation, Antithrombotic and thrombolytic activities, Hydrolyzes the Aalpha-chain of fibrinogen.
ABB79955.1	Agkihagin [*Deinagkistrodon acutus*]	0.5789	67,572.2	50	48	969	500,625.222	1.045	Binding, Enzyme_regulator_activity, Inhibits platelet aggregation. Hydrolyzes fibrinogen, Induces apoptosis and inhibits proliferation of endothelial cells.
AAF76378.1	Thrombin-like protein DAV-PA precursor [*Deinagkistrodon acutus*]	0.6008	28,031.8	21	19	133	435,941.55	0.91	Catalytic_activity, Fibrinogenolytic activities, Esterolysis and amidolytic activities.
AEJ31983.1	Phospholipase B [Crotalus adamanteus]	0.5642	64,048.6	40	20	356	375,155.62	0.78	Catalytic_activity, Hemolysis.
AAL66391.1	Antithrombin 1 A chain [*Deinagkistrodon acutus*]	0.8608	17,950.1	51	19	445	332,370.033	0.69	Platelet aggregation, Binds to human platelets, Promotes angiogenesis.
JAG44660.1	Glutaminyl-peptide cyclotransferase [Crotalus horridus]	0.5815	42,369	16	16	116	320,694.98	0.67	Binding
AAM22789.1	ACF 1/2 B-chain [*Deinagkistrodon acutus*]	0.6712	16,924.9	26	5	229	265,179.49	0.55	Binding, Anticoagulant, Inhibits platelet aggregation, Antithrombotic and thrombolytic activities, Hydrolyzes the Aalpha-chain of fibrinogen.

The protein ID is the protein number in the uniport database, and the description is the name of the protein. The protein mass indicates the relative molecular weight of the protein. snak_iBAQ represents the iBAQ value of the snake venom sample, which is used to calculate the relative content of various snake venom proteins. % Represents the relative content of snake venom protein. The functional annotations of snake venom proteins are searched through the three databases of Gene Ontology and Cluster of Orthologous Groups of proteins.

## Data Availability

The data presented in this study are available on request from the corresponding author.

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
