# Peer review of "Analysis of the Composition of Deinagkistrodon acutus Snake Venom Based on Proteomics, and Its Antithrombotic Activity and Toxicity Studies"

_molecules, 2022, doi:10.3390/molecules27072229_

Round 1

Reviewer 1 Report

Huang et al. ‘s manuscript titled ‘Analysis of the composition of Deinagkistrodon acutus snake venom based on proteomics, and its antithrombotic activity and toxicity studies’ investigated the proteins in venom via proteomic analysis from 30 different snake venom families. In addition, the pharmacological effects of snake venom show that Deinagkistrodon acutus venom may have anticoagulant and antithrombotic effects. Generally, this study is straightforward but not prepared well.

  • All the figures and tables are not presented in a professional way. For example, in Figure 1c , figure2, Figure 5B, and table 1. What is the mean for the mice figures below figure 6? Moreover, the supplemental data is even presented in Chinese.
  • The author claimed that the venom powder was purchased from a local farm but had no catalog number. Is there any evidence to show this venom powder is qualified to use in this study?
  • In 2.2.2 , …..a snake peptide database (44,716 sequences) obtained from NCBI (NCBI_sequence_snak.fasta). please provide more detailed information.
  • In 2.4, please double-check PRP or PPP.
  • The KEGG and GO analysis is also required for the identified proteins.
  • So many grammar errors throughout the manuscript.

Reviewer 2 Report

Authors analyzed Deinagkistrodon acutus snake venom by proteomics. Then, tests were performed to get an insight into the mechanism of snake venom toxicity. The authors show the correlation between the toxic effects and protein components they identified in the snake venom.

Comments

First identification of protein in D. acutus venom in Mainland China. Papers describing proteomes of D. acutus (China) have been already published (Nie X 2021, doi 10.1590/1678-9199-JVATITD-2020-0196; Huang J 2021, 10.1016/j.biopha.2021.111527).
This manuscript shows 103 proteins in D. acutus (China) venom whereas Chen 2018 published 29 proteins in D. acutus (Taiwan) venom; different proteomic methods were used. Is it possible to explain this large difference in one species only by their different geographic living localization?

Material and Methods: 
Processing of PAGE gel (lane) is not specified. It is not clear which bands were subjected to tryptic digestion (chapter 3.1, first paragraph).
Venom protein samples were filtered through 10 k ultrafiltration tubes. What was the purpose of this step?

Text
Please review the text to remove all typographic errors (e.g. Supplemental figure 1. Peptide length is in Figure B. In Abbreviations: PRP Platelet-Poor Plasma, PPP Platelet Rich Plasma) and reference numbering errors (e.g. reference 20 does not contain geographical and ontogenetic variations of toxins; reference 21 does not describe enzymatic hydrolysis.).
The purpose of the two files in Supplementary data is not clear; one of them contains figures from other papers, and the other shows SDS-PAGE assays not described in the Experimental section.

Reviewer 3 Report

Reviewer Comments:

Fang Huangand coworker present the manuscript entitled “Analysis of the composition of Deinagkistrodon acutus snake venom based on proteomics, and its antithrombotic activity and toxicity studies”. The authors presented an interesting and meaningful research. However, I have some concerns that need to be fully addressed before potential publication.

Major Points:

The main concern about this study I the lack of novelty and originality. What is the main relevance of this paper with respect to the paper reported in 2019 by Po-Chuan Chen and coworkers, named "Snake venom proteome and immuno-profiling of the hundred-pace viper, Deinagkistrodon acutus, in Taiwan"?. Authors completely ignore thi previous report on the proteomic identification of proteins from Deinagkistrodon acutus. Indeed, alarge number of proteins identified here by authors were previously identified by Po-Chuan.

In my opinion the present study is very similar to the previous Po-Chuan report, which make the manuscript lack of novelty.

A second major concern is that the authors did not perform an in-depth bioinformatics analysis of proteins, having an omics dataset. The analysis of data is very poor and very poor. The authors must use a lot of bioinformatics public software’s and also tools such as R.

Furthermore, authors must perform a comparative analysis with Po-Chuan complete data sets indicating the differences and similarities in proteins profiles.

Introduction

Some important references are missing in the introduction, such as.

Nie, X., He, Q., Zhou, B., Huang, D., Chen, J., Chen, Q., Yang, S., & Yu, X. (2021). Exploring the five-paced viper (Deinagkistrodon acutus) venom proteome by integrating a combinatorial peptide ligand library approach with shotgun LC-MS/MS. The journal of venomous animals and toxins including tropical diseases, 27, e20200196. https://doi.org/10.1590/1678-9199-JVATITD-2020-0196

Xie, C., Albulescu, L. O., Still, K., Slagboom, J., Zhao, Y., Jiang, Z., Somsen, G. W., Vonk, F. J., Casewell, N. R., & Kool, J. (2020). Varespladib Inhibits the Phospholipase A2 and Coagulopathic Activities of Venom Components from Hemotoxic Snakes. Biomedicines, 8(6), 165. https://doi.org/10.3390/biomedicines8060165

Please consider this.

The authors should rewrite this paragraph, it misses the idea “Effective separation and purification are crucial steps towards the possible clinical application of snake venom. Chromatographic separation has been a traditional method for identification of the chemical constituents of snake venom, but is unable to uncover the holistic proteome profile of the venom [4]. In addition to discovery of novel drugs, the detailed characterization of the venom proteome will not only allow us to deeply understand the pathogenesis and biological function of snake envenomation [5]. Quantification of the venom proteome was assessed applying the intensity Based Abundance Quantification (iBAQ) shotgun label-free method. With the emergence of the omics techniques, the field of proteomics has vastly expanded due to the availability of large sets of proteomic and genomic databases resulting in detailed elucidation of the composition of snake venom [6,7], the proteome profiles of the venoms of the Asiatic non-spitting cobra, N. naja atra [8] and Pakistan N. naja [9] were uncovered”. Besides, there is no link between one idea and another.

Materials and Methods

Why did you separate the proteins by SDS-PAGE at 15% and not at 10%, for example? To have a better range of proteins of both lower and higher molecular weight.

What is the difference between reducing and non-reducing conditions? And why did you use reducing conditions? Discuss your answer.

Some methodologies need to be better described, for example, Coagulant activity in vivo.

Results

Figure 1A. The authors should repeat the separation of proteins on the SDS-PAGE gel with another percentage, for example at 10% and with less amount of proteins. They do not mention the amount of proteins.

Figure 1C. The authors should redesign the image, the letters are not visible.

The authors should change Table 2 to a bubble chart or bar chart to make the GO enrichment analysis more visually understandable.

The authors should include more bioinformatics analysis from the data obtained from proteomics, for example, protein-protein infection network, pathway enrichment analysis by KEGG, among others. Bioinformatics analysis from an omics dataset is way over the top.

The authors should include well detailed figure legends, in figure 2 and 3, they are incomplete, which generates a lot of confusion.

In Figure 3A, the authors should redesign the graph, as the bar corresponding to 1.0 mg/kg is not visible.

The authors should redesign the structure of Figure 3, for example. Figure 3. A) Mortality rate (%) B) Tail (The image of the mouse tail, with its respective image of the H&E staining, and so on.

It lacks a lot of design, also in the text they mention that Figure 3A corresponds to the mortality rate and a few lines below, they mention that it corresponds to mice showing symptoms of ulceration of the tail lesion, even tail necrosis (Figures 3A).

In addition, it does not contain a figure legend.

Discussion

The authors should better discuss the results. The discussion is very brief.

Furthermore, authors must perform a comparative analysis with Po-Chuan complete data sets discussing the differences and similarities in proteins profiles.

Round 2

Reviewer 1 Report

Thanks for the revision,  the quality of the revised manuscript has been improved.

Author Response

Dear reviewer,

Reviewer 2 Report

Authors have replied all comments, I reccomend to accept the manuscript for publication in the revised form.

Author Response

Dear reviewer,

Thank you very much. Review comments and suggestions are professional and very useful to our manuscript. The manuscript of this study has been corrected in relevant sections. Thank you for the approval of our revision and valuable advice on this research.

With best regards,

Huang and Liang.

Reviewer 3 Report

Authors have succesfully replied all my concerns, this I reccomend to accept for publication the manuscript in its actual revised form.

Author Response

Dear reviewer,

Thank you very much. Review comments and suggestions are professional and very useful to our manuscript. The manuscript of this study has been corrected in relevant sections. Thank you for the approval of our revision and valuable advice on this research.

With best regards,

Huang and Liang.

This manuscript is a resubmission of an earlier submission. The following is a list of the peer review reports and author responses from that submission.

Round 1

Reviewer 1 Report

It was a very well written paper, with a detailed study of physiological activity. It was especially surprising to see the significant damage to the lungs.

Reviewer 2 Report

The article titled: “Analysis of the composition of Agkistrodon acutus snake venom based on proteomics, and its pharmacological activity and toxicity studies” the authors using proteomic approaches identified 103 proteins of the Agkistrodon acutus snake venom and related the protein profile with the toxicity and pharmacological properties of this snake venom. However, the following points should be considered.

Major revisions

  1. A previous work titled: “Transcriptome analysis of venom gland and identification of functional genes for snake venom protein in Agkistrodon acutus” Chang Sheng-xiang et al. Chinese Journal of Traditional Chinese Medicine. 2019,44 (22) should be mentioned and used to compare the similarities and differences of both investigations.
  2. Although the investigators evaluated some toxic and pharmacological effects of this snake venom, the experiments demonstrated only antithrombotic effects. In this way, the titled of the article should be focused only on these properties since toxic and pharmacological activity studies are comprised of other aspects not evaluated in this study. Furthermore, these approaches should be conducted just with protein fraction of this venom excluding other molecules (as low molecular weight components) that could interfere in the results not allowing to stablish the relation of the observed effects and the proteins identified.
  3. The discussion section must be improved, there are few relations among the results presented in the article with the literature data and with the biological effects. The limitations of the functional studies should be also pointed out specially the necessity to investigate the proteins found with proteomic approach with their functional properties.

Minor revisions

  1. Revise all text and verify the English grammar, there are some mistakes as in the first sentence of the abstract “Deinagkistrodon acutus is a strong correlation between the venom proteome and its related pharmacological effects.”
  2. Uniformize the name of the specie an put all of then in italic format - Deinagkistrodon acutus or Agkistrodon acutus.
  3. In methods section – 2.1. “Materials Preparation” it is not clear which experiments used the ICR mice and which used SPE grade SD rats. Define this in the figure’s legends also.
  4. Please define the acronyms: SPF, SPE, SD rats, PRP, ADP….
  5. In methods section – 2.4. “Platelet aggregation promoting activity” Describe the obtention of PRP. Was it obtained from health persons, blood donors? Which anticoagulant was used in the blood collection? Inform the platelet aggregation instrument used (line 118-119)
  6. Figure 1A – the SDS-Page gel electrophoresis protocol is not described in the methods section, please inform the acrylamide percentage, and describe the conditions of the electrophoresis procedures.
  7. Graphic 1C - the text is not visible put the colors and the text as legends.
  8. Review all figure legends there are missing information in all of them. Ex: Figure 1A demonstrates two images and only the SDS-page gel is informed in the figure legend…
  9. Figure 2 – Define in the figure legend the group specification – What was the blank group used?
  10. Figure 3 – In the histologic analysis are missing the scale in micrometers in each image, the amplification magnitude used (100x, 200x….). Use arrows to point the tissue alterations. Again, in the figure legend are missing information.
  11. Figure 4 – Define Blank group and control group in the figure legend.

Figure 5 – Panels A and B – There are no significance marks (*) in the panels, please revise if it is correct. There 

Reviewer 3 Report

The article “Analysis of the composition of Agkistrodon acutus snake venom based on proteomics, and its pharmacological activity and toxicity studies” analysed the composition of the venom of the Chinese Moccasin (Hundred-pacer snake) and studied by different techniques its in vivo and in vitro toxicity.

This article suffers from a number of shortcomings in both form and content.

In terms of form, the presentation is very poor. From the outset, the authors have not specified (L1) what type of paper it is: it is an article. The format does not respect the layout (the lines are too short).

The use of the English language is not mastered, and the article should be corrected by a native English speaker.

According to the IUCN red list, the Latin name of this snake is Deinagkistrodon acutus (https://www.iucnredlist.org/species/190644/1955997). Throughout the paper, the authors used both Deinagkistrodon acutus and Agkistrodon acutus, which is quite confusing. Only Deinagkistrodon acutus should be used in the article? As for all names of species, it should be written in italics.

Figure 1 is illegible and of very poor quality. The lower part of the figure does not appear.

Table 1 covers 8 pages, i.e. more than a third of the article. It is not understandable. It looks like a book of experiments. It should be completely restructured and presented in a clear and accessible way.

Figure 2 needs to be redone. The x-axis in Figure 2A shows no units and no legend. The dose of venom used is not known! The y-axis (Mortality rate) is written from top to bottom instead of bottom to top. The mortality rate has not been fitted in any way that would show a sigmoid. The equation of the curve is written on the curve, which does not allow it to be read correctly.

In this Figure 2, it is not clear why the lethality rate and % platelet aggregation are shown together, which are two very different tests.

Figure 4 has been cut off: the lower part of the figure cannot be read.

In general, the methods used in this study are poorly described, or too briefly.

L10: "In vivo and in vitro experiments showed that the LD50 of...". The determination of the LD50 can be achieved through in vivo experiments only.

Section 3.2 should be rewritten with a detailed description of the symptoms of mice exposed to different doses of venom (in a separate Table). The number of mice used per dose of venom for the lethality test should be absolutely specified. An example of the presentation of acute toxicity data is Figure 1 and Table 1 of https://doi.org/10.3390/toxins12020087. Venom toxicity data should be presented in the same way.

This article needs to be completely restructured and rewritten as it clearly cannot be published in this form.

Minor

L6: “Deinagkistrodon acutus is a strong correlation”: not clear

L8-9: “from 30 different snake venom families”. Did you mean “from 30 different snake venoms”

L12: “exhibited blood toxicity”

L13: “Within the safe dose”. What do you mean by “the safe dose” ?

L17: was performed for the first time

L20: “acutus”

L33: that snake venoms contain

L45: N. naja atra [7] and Pakistan N. naja

L47: Deinagkistrodon acutus is a medically important snake

L52: It has been reported

L101: In vitro platelet aggregation assay with Agkistrodon acutus

L127: enoxaparin sodium

L163: non-enzymatic proteins: snaclec

Figure 5: control, and not control

Etc. etc.

Reviewer 4 Report

The manuscript "Analysis of the composition of Agkistrodon acutus snake venom based on proteomics, and its pharmacological activity and toxicity studies" report a shotgun label free iBAQ-based analysis of the Chinese moccasin, a medically relevant venomous snake ranging in several South Asian countries. The authors also studied major in vivo and in vitro pathophysiological activities of the venom. The data, particularly the biological analyses, are sound and potentially interesting for the field. However, the proteomic analysis contains methodological flaws that affect the interpretation of results. My recommendation of "reject" is based on the arguments exposed below, It could also have been "Major Revision" but since reviewing the manuscript requires considerable extra work, "reject" will give authors the opportunity to perform the review without the time pressure to resubmit the revised manuscript.

1.- Zhang and colleagues from the James D. Watson Institute of Genome Sciences, Zhejiang University, Hangzhou, China; Beijing Institute of Genomics, Chinese Academy of Sciences, Beijing, China; Department of Pharmacology, Zhongshan Medical School of Sun Yat-sen University, Guangzhou, China; and the Department of Biochemistry, Zhongshan Medical School of Sun Yat-sen University, Guangzhou (China), reported in 2006 a transcriptome analysis of Deinagkistrodon acutus venom gland (Zhang B, Liu Q, Yin W, Zhang X, Huang Y, Luo Y, Qiu P, Su X, Yu J, Hu S, Yan G, Transcriptome analysis of Deinagkistrodon acutus venomous gland focusing on cellular structure and functional aspects using expressed sequence tags. BMC Genomics 2006 Jun 15;7:152. doi: 10.1186/1471-2164-7-152.). All the sequence data gathered in this study were submitted to the public database GenBank: DV556511-DV565206. The D. acutus used for the study was from Wuyi Mountain, Fujian Province, China.

Ten years later, a consortium of Chinese researchers, including some of the authors of the first transcriptome analysis of Deinagkistrodon acutus venom gland, reported the 1.43 Gb genome and tissue transcriptomes (including the venom gland) of a male and a female of the five-pacer viper, Deinagkistrodon acutus, sequenced to high-coverage (female, ~ 238 fold; male, 114 fold) from blood and eight adult tissues (Yin W, Wang ZJ, Li QY, Lian JM, Zhou Y, Lu BZ, Jin LJ, Qiu PX, Zhang P, Zhu WB, Wen B, Huang YJ, Lin ZL, Qiu BT, Su XW, Yang HM, Zhang GJ, Yan GM, Zhou Q, Evolutionary trajectories of snake genes and genomes revealed by comparative analyses of five-pacer viper. Nat Commun. 2016 Oct 6;7:13107. doi: 10.1038/ncomms13107). The authors of this study annotated a total of 35 venom genes or gene families using all the known snake venom proteins as the query.

The genomic and RNA-seq reads were deposited with the NCBI Short Reads Archive under the BioProject Accession Number PRJNA314443, and BioProject Accession Number PRJNA314559, respectively. The genome assembly and annotation are deposited in the GigaScience Database http://dx.doi.org/10.5524/100196.

However, none of these seminal studies are cited by the authors of the current paper submitted to Toxins. Thus, the authors claim that " The present study addresses, for the first time, the composition and content of protein peptides in Agkistrodon acutus venom originating in Chinese mainland" is not accurate.

2.- The venom proteome analysis is rather poorly described. Details of the analytical workflows applied; false discovery rate and search database engine employed; strategy and statistics of protein identification and quantification followed, should be provided. Furthermore, the raw mass spectrometric dataset(s) should be deposited with a public accessible repository, such as ProteomXchange via PRIDE, and the username and password for peer review should be provided.

3.- Table 1 lists the assignments of Deinagkistrodon acutus venom proteome. Table 1, or a Supplementary Table, should show the MS details (e.g., m/z, z, amino acid sequence, ID score, best database match, etc.) of the precursor ions used for the assignments. Further, the column "Function" is vague and does not add anything that aids in the characterization of the proteins.

4.- Quantification of the venom proteome was assessed applying the intensity Based Abundance Quantification (iBAQ) shotgun label-free method. Label-free methods can conveniently provide comparative estimates of peptide levels in multiplex differential proteomics experiments. Accurate quantification of the components of a proteome in a single experiment is only consistent if the identification of the peptides is carried out against a comprehensive homolog database. The strategy of label-free quantification by spectral counting or ion intensity is based on the assumption that the likelihood of data-dependent precursor ion selection is higher for abundant precursor ions, and that the number of peptide identifications, normalised to account for the fact that larger proteins tend to contribute more peptide/spectra, represents a proxy of the abundance of the parent protein. Central to label-free quantification methods is the ratio "Number of observed peptides/Number of theoretical observable peptides. Missing values is one of the main problems associated with label-free proteomics. Label-free proteome quantification strategies have been developed for model organisms for which comprehensive genomic or transcriptomic databases are available, and thus protein identifications does not represent a limiting factor. The conceptual shortcoming of applying label free shotgun approaches to non-model organisms results from the fact when a comprehensive reference database is missing, quantification is biased toward the successful peptide identifications. The authors are, thus, advised to carry out a repetition of the venom proteome quantification matching the MS/MS data against the species-specific genomic/transcriptomic database.

            On the other hand, intensity-based absolute quantitation (iBAQ) is derived from the sum of all peptide peaks for a given protein divided by the number of theoretically observed peptides. However, mass spectrometry is inherently a not quantitative technique due to numerous confounding factors contributing to the detection of peptide ions in a mass spectrometer. Most notably, the distinct physico-chemical sequence properties of the different analytes in any given sample unpredictably affect peptide ionization, and detection efficiencies for ions with different m/z values are unequal. Hence, quantification of the venom proteome of D. acutus should be derived from data gathered through a truly quantitative method, such as snake venomics (). Snake venomics leverages on pre-MS quantification of the percentage of the different toxin families in the venom from the ratio of the summed areas of the reverse-phase chromatographic peaks containing proteins from the same family to the total area of venom protein peaks in the RP chromatogram recorded at the wavelength of absorbance for the peptide bond (Calvete JJ, Next-generation snake venomics: protein-locus resolution through venom proteome decomplexation.Exp. Rev. Proteomics 2014 Jun;11(3):315-29. doi: 10.1586/14789450.2014.900447; Eichberg S, Sanz L, Calvete JJ, Pla D, Constructing comprehensive venom proteome reference maps for integrative venomics. Exp. Rev. Proteomics. 2015;12(5):557-73. doi: 10.1586/14789450.2015.1073590). These percentages correspond to the “% of total peptide bond concentration in the peak”, a proxy for the weight% (Calvete JJ, Pla D, Els J, Carranza S, Damm M, Hempel BF, John EBO, Petras D, Heiss P, Nalbantsoy A, Göçmen B, Süssmuth RD, Calderón-Celis F, Nosti AJ, Encinar JR, Combined Molecular and Elemental Mass Spectrometry Approaches for Absolute Quantification of Proteomes: Application to the Venomics Characterization of the Two Species of Desert Black Cobras, Walterinnesia aegyptia and Walterinnesia morgani. J. Proteome Res. 2021 Nov 5;20(11):5064-5078. doi: 10.1021/acs.jproteome.1c00608.).  

5.- A comparative analysis of the current and previous omics data available for D. acutus from PR China and Taiwan should be carry out and discussed.